# Engaging Patients in Smoking Cessation Treatment within the Lung Cancer Screening Setting: Lessons Learned from an NCI SCALE Trial

Randi M. Williams [1],*, Ellie Eyestone [1], Laney Smith [1], Joanna G. Philips [1], Julia Whealan [1], Marguerite Webster [1], Tengfei Li [2], George Luta [2], Kathryn L. Taylor [1] and on behalf of the Lung Screening, Tobacco, Health Trial [†]

[1] Cancer Prevention and Control Program, Lombardi Comprehensive Cancer Center, Georgetown University Medical Center, Washington, DC 20007, USA; ee292@georgetown.edu (E.E.); ls1434@georgetown.edu (L.S.); jp1878@georgetown.edu (J.G.P.); jw2057@georgetown.edu (J.W.); maw342@georgetown.edu (M.W.); taylorkl@georgetown.edu (K.L.T.)

[2] Department of Biostatistics, Bioinformatics and Biomathematics, Georgetown University, Washington, DC 20007, USA; tl602@georgetown.edu (T.L.); george.luta@georgetown.edu (G.L.)

* Correspondence: rmw27@georgetown.edu; Tel.: +1-202-687-7036

† Collaborators indicated in the Acknowledgment section.

**Abstract:** Offering smoking cessation treatment at lung cancer screening (LCS) will maximize mortality reduction associated with screening, but predictors of treatment engagement are not well understood. We examined participant characteristics of engagement in an NCI SCALE cessation trial. Eligible LCS patients (N = 818) were randomized to the Intensive arm (8 phone counseling sessions +8 weeks of nicotine replacement therapy (NRT)) vs. Minimal arm (3 sessions + 2 weeks of NRT). Engagement was measured by number of sessions completed (none, some, or all) and NRT mailed (none vs. any) in each arm. In the Intensive arm, those with ≥some college (OR = 2.1, 95% CI = 1.1, 4.0) and undergoing an annual scan (OR = 2.1, 95% CI = 1.1, 4.2) engaged in some counseling vs. none. Individuals with higher nicotine dependence were more likely (OR = 2.8, 95% CI = 1.3, 6.2) to request NRT. In the Minimal arm, those with higher education (OR = 2.1, 95% CI = 1.1, 3.9) and undergoing an annual scan (OR = 2.0, 95% CI = 1.04, 3.8) completed some sessions vs. none. Requesting NRT was associated with more pack-years (OR = 1.9, 95% CI = 1.1, 3.5). Regardless of treatment intensity, additional strategies are needed to engage those with lower education, less intensive smoking histories, and undergoing a first scan. These efforts will be important given the broader 2021 LCS guidelines.

**Keywords:** tobacco treatment; smoking cessation; lung cancer screening; intervention engagement

## 1. Introduction

Providing evidence-based smoking cessation interventions at the time of low-dose computed tomographic lung cancer screening (lung screening) is estimated to substantially reduce lung cancer deaths and increase life-years gained compared to conducting lung screening alone [1]. However, despite the effectiveness of the cessation treatment services accompanying lung screening programs, lower reach, enrollment, and/or engagement will limit the overall impact that lung screening will have on health outcomes. Based on the classic reach x effectiveness = impact paradigm [2], regardless of the effectiveness achieved, low reach will limit the impact that treatment will have on the number who quit, and ultimately, on mortality reduction [3]. In the current paper, we have assessed the factors related to reach and engagement among individuals offered cessation services in the lung screening context with the goal of maximizing the public health benefit of pairing smoking cessation treatment with lung screening [4–6].

### 1.1. Predictors of Enrollment and Retention in Smoking Cessation Trials in the Lung Screening Setting

To determine how best to integrate smoking cessation treatment in the lung screening setting, the National Cancer Institute initiated the Smoking Cessation at Lung Examination (SCALE) collaboration [7] that is comprised of eight clinical trials, including Georgetown's Lung Screening, Tobacco, and Health (LSTH) trial (NCT03200236) [8–12]. To date, several SCALE-related papers have assessed predictors of enrollment and retention [7,10,13]. In a cross-project analysis of enrollment across six of the eight SCALE trials (*n* = 6285), we found that participants were more likely to be Black or African American, were less likely to be of Hispanic ethnicity, and were significantly younger compared to their counterparts among those who declined enrollment or could not be reached [7]. The offer of multimodal counseling along with nicotine replacement therapy (NRT) plus prescription medication were strong predictors of enrollment compared to trials offering phone-only counseling or NRT alone. In addition to this pooled analysis, the Massachusetts General Hospital trial (NCT03611881) found that intent to enroll in a smoking cessation study was greater among individuals who placed a higher importance on quitting, felt the recruitment messages were more relevant to them, and were more worried about developing lung cancer [10]. The Georgetown LSTH trial found that attrition between the baseline assessment and the post-lung screening assessment was more likely among individuals who were younger, with lower education, Hispanic, who received a negative lung screening result, reported little worry about lung cancer, and were undergoing their first lung screening exam [13]. While other cessation trials in the lung screening setting have been conducted, few have provided details on the predictors of trial enrollment and retention [4,5,14–17].

### 1.2. Predictors of Intervention Engagement in Smoking Cessation Trials

Although the literature describing predictors of intervention engagement in SCALE trials is limited to date, there has been substantial research in the area of engagement and adherence to cessation programs and quitlines in other settings [6,18–21]. The current literature on this topic suggests that women [6]; older adults [18,19]; and individuals with higher income [20], higher education [20], and health insurance [20] have greater treatment engagement. Further, higher nicotine dependence [6,18] and higher motivation to quit [20] are associated with higher treatment engagement and adherence, although there have been exceptions to these findings [22]. Types of cessation support provided were also related to engagement, suggesting that provision of pharmacotherapy is associated with greater engagement and adherence [18,21]. While these studies can help inform barriers and facilitators to engagement in smoking cessation programs, individuals undergoing lung screening differ from the general population of individuals who smoke on gender, age, and motivation to quit [22–24], highlighting the need to understand predictors of treatment engagement in this context.

In this secondary data analysis, we examined predictors of engagement in our multi-site smoking cessation randomized clinical trial, evaluating two telephone counseling interventions with NRT. Additionally, we have described the lessons learned around reach and engagement that can inform successful integration of tobacco treatment in the lung screening setting.

## 2. Materials and Methods

### 2.1. Participants

To be eligible for the study, participants met the National Comprehensive Cancer Network Group 1 and/or Group 2 guidelines for LCS: age 50–80 years old, a smoking history of at least 20 pack-years, and >one additional risk factor. Additional inclusion criteria were: (1) smoked within the past 7 days at enrollment, (2) English or Spanish-speaker, and (3) had registered for but not yet completed a lung screening scan. Individuals were not excluded from the study based on having completed a prior lung cancer screening exam, previous or ongoing cessation treatment, readiness to quit, or psychiatric or comorbid conditions.

## 2.2. Procedures

Details about the study design and methods are published elsewhere [8]. Patients scheduled for lung screening at 8 screening sites were approached for enrollment in the LSTH trial between May 2017 and January 2021. The screening sites were located in geographically diverse community-based hospitals and academic medical centers. Individuals who had scheduled a screening appointment were approached by site study coordinators who described the study to eligible individuals, obtained verbal consent, and conducted a baseline telephone assessment (T0) prior to their lung screening CT scan. After completing the scan and receiving the results, tobacco treatment specialists (TTS) from Georgetown University Medical Center called participants to administer the post-screening assessment (T1) and randomize participants (*n* = 818) to one of two study arms. The Intensive arm provided up to 8 telephone counseling sessions (20 min each) and up to 8 weeks of nicotine patches (NicoDerm CQ 21 mg, 14 mg, and 7 mg), hereafter referred to as NRT (*n* = 409). The Minimal arm provided up to 3 telephone counseling sessions (20 min each) and 2 weeks of NRT (*n* = 409). Follow-up assessments were conducted at 3-, 6-, and 12-months post-randomization. The study was approved by the Georgetown University Medical Center Institutional Review Board (IRB) (IRB of Record) and by the Lahey Hospital and Medical Center IRB.

After completing the post-screening assessment, participants were mailed or emailed intervention materials for use during counseling sessions. Participants had three months to complete the intervention and were required to complete at least one counseling session prior to receiving the NRT. Participants in the Minimal arm were approached up to three times for each counseling session while participants in the Intensive arm were approached up to six times for each session.

Telephone counseling was conducted by TTSs who were trained and supervised on protocol adherence and in motivational interviewing (MI) [25–28]. Counseling calls were guided by a protocol that included MI-based discussions of motivation and confidence to quit, individual barriers and goals toward cutting down or quitting smoking, and readiness to quit. For participants in the Intensive arm only, lung screening results and lung screening as a possible teachable moment [29–32] were discussed during sessions 1–3. We coded a random selection of 10% of the counseling calls on 25 items measuring protocol adherence. The TTSs maintained high adherence to the protocol: M = 94.5% (88–100) in the Intensive arm and M = 95.5% (89–100) in the Minimal arm. [33] Inter-rater reliability was calculated for 20% of the coded calls and showed high overall agreement for each category: M = 95.0% (80–100). The counseling protocol is available upon request.

## 2.3. Measures

All measures described were assessed prior to randomization except the intervention feedback variables that were assessed at the assessment that followed completion of the intervention.

Demographics and Clinical Information. We collected self-reported age, gender, race, marital status, and education. More than 95% of the sample was insured and so we did not include this variable in the analysis. Results of participants' lung cancer scans were obtained using the electronic health record (EHR). Lung screening results were communicated using the Lung-RADS® system (American College of Radiology Committee on Lung-RADS®): Lung-RADS® 0 = incomplete; Lung-RADS® 1 = negative result; Lung-RADS® 2 = benign appearance or behavior; Lung-RADS® 3 = probably benign finding(s); and Lung-RADS® 4 = suspicious with recommended follow-up specialist consultation, imaging and/or diagnostic procedures. For the present analysis, we collapsed Lung-RADS® into a 2-level variable (negative: Lung-RADS® 1, 2 and positive: Lung-RADS® 3, 4). We also assessed whether participants were undergoing their first (baseline) vs. an annual repeat lung screening via the EHR.

Tobacco-related Characteristics. We collected tobacco-related information including cigarettes per day (CPD), history of cigarette smoking (pack-years), time to first cigarette,

readiness to quit [34] (plan on quitting in the next 30 days; plan on quitting in the next 6 months; not considering quitting), attitudes toward quitting (motivation to quit and confidence in quitting: 1 (low) to 10 (high)), and previous evidence-based cessation methods utilized (i.e., nicotine replacement gum, lozenges, patches; prescription medications; group, individual in-person or telephone counseling).

Psychological Characteristics. We assessed perceived lung cancer risk (lower, about the same, higher risk than other individuals who smoke) [35] and worry about lung cancer (no/little worry, somewhat, extremely worried) [30].

Outcome Engagement Variables. To measure intervention engagement, we used two variables: telephone counseling engagement was based upon sessions completed, with no engagement defined as 0 sessions (both arms), some engagement as 1–7 sessions completed in the Intensive arm and 1–2 sessions completed in the Minimal arm, and complete engagement as 8 sessions in the Intensive and 3 sessions in the Minimal arm. All participants were offered NRT, and we mailed it to those who requested it. We defined NRT engagement as none (no NRT mailed) or some (at least 2 weeks of NRT mailed in both arms). We compared the NRT mailed variable to the item asked at follow-up ('Did you receive nicotine patches from our project?') and found the responses were very similar (98.5% agreement). We used the NRT mailed variable as it was more complete and was not limited to those who completed a follow-up assessment.

Intervention Feedback Variables. We collected participant feedback regarding the interventions during the first follow-up assessment in which a participant was reached. Participants rated their use of and satisfaction with the counseling sessions and NRT offered by the trial.

### 2.4. Statistical Analysis

We examined predictors of engagement within each study arm. Chi-square tests and ANOVAs were used to examine associations between the treatment engagement outcomes and the demographic, clinical, tobacco-related, and psychological variables. Analyses that included the lung screening site variable resulted in small cell sizes and were not included in multivariable analyses. Variables associated with the engagement outcomes ($p < 0.20$) were included in the multivariable models. We conducted two separate multinomial logistic regression models (Intensive arm, Minimal arm) to examine the demographic, clinical, smoking-related, and psychological predictors of the 3-level counseling engagement outcome variable (none, some, and complete engagement). Similarly, we conducted two separate logistic regression models for each study arm to assess predictors of NRT engagement (none or some NRT mailed). We examined different cut points for measuring counseling engagement, including a 2-level variable. Due to the distribution of the data, including the proportion of participants who completed either all or none of the sessions, we were concerned that the 2-level variables might obscure associations with the predictors of interest and thus elected to use the 3-level variable. Finally, we describe the participants' feedback on the interventions. All data were analyzed using SPSS version 28.

### 3. Results

Figure 1a,b presents the percentage of participants who engaged in counseling and/or received NRT in the Intensive and Minimal intervention arms, respectively. In the Intensive arm, 37.9% completed all 8 sessions, and 81.4% received between 2–8 weeks of NRT. In the Minimal arm, 51.1% completed all 3 counseling sessions, and 73.1% received 2 weeks of NRT.

### 3.1. Univariate Predictors of Engagement in the Intensive Arm

The background and clinical characteristics stratified by engagement are presented in Table 1a (counseling) and b (NRT). Predictors of complete counseling engagement (8 sessions) were being male and white (*p*'s < 0.20). Predictors of 'no engagement' in counseling were undergoing a baseline scan and having lower education. Requesting NRT

was associated with higher education, higher nicotine dependence, and higher motivation to quit.

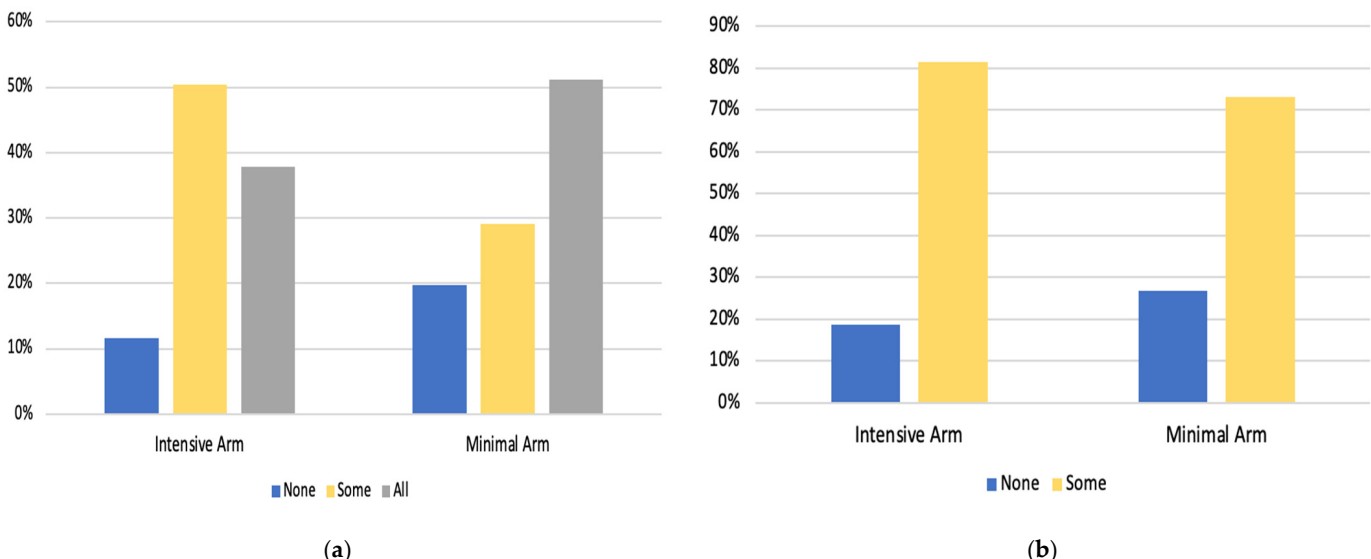

(**a**) (**b**)

**Figure 1.** (**a**). Counseling engagement by study arm. Note: For the Intensive arm: none (0 sessions), some (1–7 sessions), and all (8 sessions). For the Minimal arm: none (0 sessions), some (1–2 sessions), and all (3 sessions). (**b**). Nicotine replacement therapy engagement by study arm. Note: For the Intensive arm: none (no NRT) and some NRT (2 to 8 weeks of NRT). For the Minimal arm: none (no NRT) and some (2 weeks of NRT).

**Table 1.** (**a**) Characteristics by counseling engagement. (**b**) Characteristics by NRT engagement.

| (a) | | | | | | |
|---|---|---|---|---|---|---|
| | **Counseling Engagement** | | | | | |
| | **Intensive Arm** **(n = 409)** | | | **Minimal Arm** **(n = 409)** | | |
| | **No Engagement (0 Sessions)** **n = 48 (11.7%)** | **Some Engagement (1–7 Sessions)** **n = 206 (50.4%)** | **Complete Engagement (8 Sessions)** **n = 155 (37.9%)** | **No Engagement (0 Sessions)** **n = 81 (19.8%)** | **Some Engagement (1–2 Sessions)** **n = 119 (29.1%)** | **Complete Engagement (3 Sessions)** **n = 209 (51.1%)** |
| | Demographics | | | | | |
| Age, *n* (%) | | | | | | |
| 50–63 Years | 25 (11.8) | 115 (54.2) | 72 (34.0) | 51 (23.7) | 66 (30.7) | 98 (45.6) *** |
| 64–80 Years | 23 (11.7) | 91 (46.2) | 83 (42.1) | 30 (15.5) | 53 (27.3) | 111 (57.2) |
| Gender, *n* (%) | | | | | | |
| Male | 21 (10.7) | 89 (45.2) | 87 (44.2) *** | 35 (18.3) | 52 (27.2) | 104 (54.5) |
| Female | 27 (12.7) | 117 (55.2) | 68 (32.1) | 46 (21.1) | 6 (30.7) | 105 (48.2) |
| Race, *n* (%) | | | | | | |
| White | 41 (11.3) | 179 (49.2) | 144 (39.6) * | 70 (19.2) | 110 (30.1) | 185 (50.7) |
| Black | 6 (17.6) | 20 (58.8) | 8 (23.5) | 7 (22.6) | 7 (22.6) | 17 (54.8) |
| Marital Status, *n* (%) | | | | | | |
| Married or in Marriage-like Relationship | 24 (11.6) | 100 (48.3) | 83 (40.1) | 35 (17.7) | 70 (35.4) | 93 (47.0) *** |
| Not Married | 24 (11.9) | 105 (52.2) | 72 (35.8) | 45 (21.5) | 4 (23.4) | 115 (55.0) |
| Education Level, *n* (%) | | | | | | |
| High School/GED or Less | 24 (16.8) | 65 (45.5) | 54 (37.8) ** | 38 (26.6) | 37 (25.9) | 68 (47.6) *** |
| Some College or Greater | 24 (9.1) | 140 (53.0) | 100 (37.9) | 42 (16.0) | 82 (31.2) | 139 (52.9) |
| | Clinical Characteristics | | | | | |
| Lung Cancer Screening Result, *n* (%) | | | | | | |
| Lung-RADS 1/2 | 45 (122) | 187 (50.5) | 138 (37.3) | 72 (19.7) | 108 (29.65) | 186 (50.8) |
| Lung-RADS 3/4 | 3 (7.7) | 19 (48.7) | 17 (43.6) | 9 (20.9) | 11 (25.6) | 23 (53.5) |
| LDCT Screening, *n* (%) | | | | | | |
| Baseline scan | 29 (16.5) | 85 (48.3) | 62 (35.2) *** | 40 (23.7) | 43 (25.4) | 86 (50.9) * |
| Annual Scan | 19 (8.2) | 121 (51.9) | 93 (39.9) | 41 (17.1) | 76 (31.7) | 123 (51.3) |

**Table 1.** *Cont.*

**(a)**

| | Counseling Engagement | | | | | |
|---|---|---|---|---|---|---|
| | Intensive Arm (*n* = 409) | | | Minimal Arm (*n* = 409) | | |
| | No Engagement (0 Sessions) *n* = 48 (11.7%) | Some Engagement (1–7 Sessions) *n* = 206 (50.4%) | Complete Engagement (8 Sessions) *n* = 155 (37.9%) | No Engagement (0 Sessions) *n* = 81 (19.8%) | Some Engagement (1–2 Sessions) *n* = 119 (29.1%) | Complete Engagement (3 Sessions) *n* = 209 (51.1%) |
| Tobacco-Related Characteristics | | | | | | |
| Pack Years, *n* (%) | | | | | | |
| 20–39 | 13 (11.2) | 62 (53.4) | 41 (35.3) | 26 (21.7) | 38 (31.7) | 56 (46.7) |
| 40–49 | 21 (13.5) | 75 (48.4) | 59 (38.1) | 33 (21.2) | 47 (30.1) | 76 (48.7) |
| 50+ | 14 (10.1) | 69 (50.0) | 55 (39.9) | 22 (16.8) | 33 (25.2) | 76 (58.0) |
| Cigarettes per day, *n* (%) | | | | | | |
| Less than 20 | 26 (11.7) | 109 (49.1) | 87 (39.2) | 46 (21.5) | 57 (26.6) | 111 (51.9) |
| 20 or more | 21 (11.4) | 95 (51.6) | 68 (37.0) | 35 (18.0) | 62 (32.0) | 97 (50.0) |
| Time to First Cigarette, *n* (%) | | | | | | |
| Within 5 min | 17 (14.2) | 55 (45.8) | 48 (40.0) | 30 (23.8) | 34 (27.0) | 62 (49.2) *** |
| 6 to 30 min | 14 (8.4) | 87 (52.4) | 65 (39.2) | 35 (20.6) | 46 (27.1) | 89 (52.4) |
| 31 to 60 min | 8 (11.1) | 37 (51.4) | 27 (37.5) | 5 (9.3) | 26 (48.1) | 23 (42.6) |
| After 60 min | 7 (15.2) | 25 (54.3) | 14 (30.4) | 10 (18.2) | 12 (21.8) | 33 (60.0) |
| Readiness to Quit, *n* (%) | | | | | | |
| Not considering quitting | 20 (15.3) | 66 (50.4) | 45 (34.4) | 25 (19.1) | 38 (29) | 68 (51.9) |
| Next 6 months | 8 (10.3) | 42 (53.8) | 28 (35.9) | 22 (26.8) | 26 (31.7) | 34 (41.5) |
| Next 30 days | 20 (10) | 98 (49) | 82 (41) | 34 (17.3) | 55 (28.1) | 107 (54.6) |
| Motivation to Quit, Mean (SD), Median (1 = low motivation, 10 = high motivation) | 6.30 (2.5), 6.00 | 6.83 (2.3), 7.00 | 6.76 (2.3), 7.00 | 6.38 (2.4), 6.00 | 6.64 (2.0), 7.00 | 6.77 (2.3), 7.00 |
| Confidence in Quitting, Mean (SD), Median (1 = low confidence, 10 = high confidence) | 5.66 (2.7), 6.00 | 5.81 (2.4), 5.00 | 6.03 (2.5), 6.00 | 5.62 (2.5), 5.00 | 5.53 (2.7), 5.00 | 6.05 (2.6), 6.00 * |
| Psychological Variables | | | | | | |
| Comparative Risk, *n* (%) | | | | | | |
| Lower risk | 7 (11.1) | 29 (46.0) | 27 (42.9) | 18 (29.0) | 19 (30.6) | 25 (40.3) * |
| About the same | 18 (10.8) | 82 (49.4) | 66 (39.8) | 33 (20.8) | 41 (25.8) | 85 (53.5) |
| Higher risk | 18 (11.0) | 88 (54.0) | 57 (35.0) | 25 (15.3) | 52 (31.9) | 86 (52.8) |
| Worry about Lung Cancer, *n* (%) | | | | | | |
| No/Little Worry | 11 (11.7) | 50 (53.2) | 33 (35.1) | 16 (17.8) | 28 (31.1) | 46 (51.1) |
| Somewhat | 20 (12.3) | 78 (47.9) | 65 (39.9) | 26 (16.1) | 51 (31.7) | 84 (52.2) |
| Extremely | 16 (10.8) | 77 (52.0) | 55 (37.2) | 36 (24.2) | 39 (26.2) | 74 (49.7) |

**(b)**

| | Nicotine Replacement Therapy (NRT) Engagement | | | |
|---|---|---|---|---|
| | Intensive Arm | | Minimal Arm | |
| | (*n* = 409) | | (*n* = 409) | |
| | No NRT *n* = 76 (18.6%) | Some NRT (1–4 Boxes) *n* = 333 (81.4%) | No NRT *n* = 110 (26.9%) | Some NRT (1 Box) *n* = 299 (73.1%) |
| Demographics | | | | |
| Age, *n* (%) | | | | |
| 50–63 Years | 39 (18.4) | 173 (81.6) | 61 (28.4) | 154 (71.6) |
| 64–80 Years | 37 (18.8) | 160 (81.2) | 49 (25.3) | 145 (74.7) |
| Gender, *n* (%) | | | | |
| Male | 33 (16.8) | 164 (83.2) | 47 (24.6) | 144 (75.4) |
| Female | 43 (20.3) | 169 (79.7) | 63 (28.9) | 155 (71.1) |
| Race, *n* (%) | | | | |
| White | 67 (18.4) | 297 (81.6) | 97 (26.6) | 268 (73.4) |
| Black | 6 (17.6) | 28 (82.4) | 9 (29.0) | 22 (71.0) |
| Marital Status, *n* (%) | | | | |
| Married or in Marriage-like Relationship | 39 (18.8) | 168 (81.2) | 51 (25.8) | 147 (74.2) |
| Not Married | 37 (18.4) | 164 (81.6) | 58 (27.8) | 151 (72.2) |
| Education Level, *n* (%) | | | | |
| High School/GED or Less | 32 (22.4) | 111 (77.6) * | 42 (29.4) | 101 (70.6) |
| Some College or Greater | 44 (16.7) | 220 (83.3) | 67 (25.5) | 196 (74.5) |
| Clinical Characteristics | | | | |
| Lung Cancer Screening Result, *n* (%) | | | | |
| Lung-RADS 1/2 | 71 (19.2) | 299 (80.8) | 98 (26.8) | 268 (73.2) |
| Lung-RADS 3/4 | 5 (12.8) | 34 (87.2) | 12 (27.9) | 31 (72.1) |
| LDCT Screening, *n* (%) | | | | |
| Baseline scan | 36 (20.5) | 140 (79.5) | 48 (28.4) | 121 (71.6) |
| Annual Scan | 40 (17.2) | 193 (82.8) | 62 (25.8) | 178 (74.2) |

**Table 1.** *Cont.*

| | | | | |
|---|---|---|---|---|
| **(b)** | | | | |
| | **Nicotine Replacement Therapy (NRT) Engagement** | | | |
| | **Intensive Arm** | | **Minimal Arm** | |
| | (*n* = 409) | | (*n* = 409) | |
| | **No NRT *n* = 76 (18.6%)** | **Some NRT (1–4 Boxes) *n* = 333 (81.4%)** | **No NRT *n* = 110 (26.9%)** | **Some NRT (1 Box) *n* = 299 (73.1%)** |
| **Tobacco-Related Characteristics** | | | | |
| Pack Years, *n* (%) | | | | |
|   20–39 | 19 (16.4) | 97 (83.6) | 41 (34.2) | 79 (65.8) ** |
|   40–49 | 34 (21.9) | 121 (78.1) | 40 (25.6) | 116 (74.4) |
|   50+ | 23 (16.7) | 115 (83.3) | 29 (22.1) | 102 (77.9) |
| Cigarettes per day, *n* (%) | | | | |
|   Less than 20 | 42 (18.9) | 180 (81.1) | 65 (30.4) | 149 (69.6) * |
|   20 or more | 33 (17.9) | 151 (82.1) | 45 (23.2) | 149 (76.8) |
| Time to First Cigarette, *n* (%) | | | | |
|   Within 5 min | 25 (20.8) | 95 (79.2) ** | 35 (27.8) | 91 (72.2) |
|   6 to 30 min | 22 (13.3) | 144 (86.7) | 45 (26.5) | 125 (73.5) |
|   31 to 60 min | 13 (18.1) | 59 (81.9) | 10 (18.5) | 44 (81.5) |
|   After 60 min | 13 (28.3) | 33 (71.7) | 17 (30.9) | 38 (69.1) |
| Readiness to Quit, *n* (%) | | | | |
|   Not considering quitting | 30 (22.9) | 101 (77.1) | 37 (28.2) | 94 (71.8) |
|   Next 6 months | 12 (15.4) | 66 (84.6) | 24 (29.3) | 58 (70.7) |
|   Next 30 days | 34 (17.0) | 166 (83.0) | 49 (25.0) | 147 (75.0) |
| Motivation to Quit, Mean (SD), Median (1 = low motivation, 10 = high motivation) | 6.38 (2.5), 6.00 | 6.82 (2.3), 7.00 * | 6.47 (2.4), 6.00 | 6.73 (2.2), 7.00 |
| Confidence in Quitting, Mean (SD), Median (1 = low confidence, 10 = high confidence) | 5.63 (2.6), 5.00 | 5.93 (2.5), 6.00 | 5.80 (2.6), 6.00 | 5.82 (2.6), 6.00 |
| **Psychological Variables** | | | | |
| Comparative Risk, *n* (%) | | | | |
|   Lower risk | 12 (19.0) | 51 (81.0) | 24 (38.7) | 38 (61.3) *** |
|   About the same | 30 (18.1) | 136 (81.9) | 43 (27.0) | 116 (73.0) |
|   Higher risk | 29 (17.8) | 134 (82.2) | 32 (19.6) | 131 (80.4) |
| Worry about Lung Cancer, *n* (%) | | | | |
|   No/Little Worry | 16 (17.0) | 78 (83.0) | 26 (28.9) | 64 (71.1) |
|   Somewhat | 33 (20.2) | 130 (79.8) | 36 (22.4) | 125 (77.6) |
|   Extremely | 25 (16.9) | 123 (83.1) | 43 (28.9) | 106 (71.1) |

\* $p < 0.20$; ** $p < 0.10$; *** $p < 0.05$. Note: Previous cessation methods used were not significantly associated with the engagement outcome variables.

### 3.2. Univariate Predictors of Engagement in the Minimal Arm

Predictors of complete engagement in counseling in the Minimal arm (Table 1a) included being older, not being married, having lower nicotine dependence, and having greater confidence in quitting ($p$'s < 0.20). Similar to the Intensive arm, those coming in for a baseline scan, with lower education or lower perceived lung cancer risk completed fewer sessions. Predictors of requesting NRT (Table 1b) were more CPD, more pack-years, and higher perceived risk ($p$'s < 0.20).

### 3.3. Regression Models Predicting Engagement in the Intensive Arm

In multivariable analyses, in the Intensive arm, those with higher education (some college) vs. a high school degree or less (OR = 2.1. 95% CI = 1.1, 4.0) and those coming in for an annual scan vs. a baseline scan (OR = 2.1, 95% CI = 1.1, 4.2) had greater odds of engaging in some counseling compared to no sessions. Requesting NRT was greater among individuals with higher nicotine dependence (6–30 min from waking; OR = 2.8, 95% CI = 1.3, 6.2) vs. greater than 60 min from waking (Table 2a,b).

### 3.4. Regression Models Predicting Engagement in the Minimal Arm

In the Minimal arm, those with higher education were more likely to complete some (OR = 2.1, 95% CI = 1.1, 3.9) or all sessions (OR = 1.9, 95% CI = 1.1, 3.4) compared to no sessions. Those coming in for an annual scan (OR = 2.0, 95% CI = 1.04, 3.8), with higher addiction (6–30 min from waking; OR = 3.2, 95% CI = 1.03, 9.9), and those who were married (OR = 2.0, 95% = 1.1, 3.8) were more likely to engage in some sessions vs. no sessions. Older individuals (OR = 1.9, 95% CI = 1.1, 3.4) and those with high perceived

risk (OR = 2.8, 95% CI = 1.2, 6.6) were more likely to complete all 3 sessions compared to no counseling engagement. Requesting NRT was associated with higher perceived risk (OR = 2.7, 95% CI = 1.4, 5.2) and 50+ pack years (OR = 1.9, 95% CI = 1.1, 3.5) (Table 2a,b).

### 3.5. Intervention Feedback Variables

Of those who completed the feedback survey (*n* = 618, 90% of those who completed >1 counseling session), 95.5% of the Intensive arm participants reported receiving telephone counseling support, whereas 89.8% of the Minimal arm participants reported receiving counseling cessation support (*p* < 0.01). Intensive arm participants were more likely to report being very satisfied with counseling (84%) compared to the Minimal arm (68.1%; *p* < 0.001). The majority of Intensive and Minimal arm participants reported receiving NRT from the LSTH trial (87.7% vs. 82.2%, respectively). Regarding satisfaction with the NRT received, the Intensive arm was significantly more satisfied compared to the Minimal arm (58.4% vs. 45.7% were very satisfied, respectively; *p* < 0.01). Among those who reported receiving telephone counseling, the Intensive arm participants were more likely to respond 'very much' when asked if the counseling provided by the LSTH project helped them to become more ready to quit smoking vs. the Minimal arm (70.4% and 53.8%, respectively; *p* < 0.001). Similarly, regarding whether the NRT helped participants to become more ready to quit, Intensive arm participants were more likely to indicate "very much" compared to Minimal arm participants (55.6% vs. 36.5%, respectively; *p* < 0.001). Minimal arm participants were more likely to purchase more NRT compared to the Intensive group (*p* < 0.001). When asked if they would have preferred more counseling calls, fewer calls, or if it was the right number of calls, individuals in the Intensive arm were more likely to indicate they were offered the right number of calls (71.5%) compared to the Minimal arm (59.8%; *p* = 0.001). There was no difference by study arm of preference for modality of cessation counseling: 46.8% preferred telephone counseling, 45.1% had no preference, and 8.1% preferred in-person counseling sessions (Table 3).

**Table 2.** (**a**) Multinomial logistic regression models predicting counseling engagement (*n* = 818). (**b**) Multinomial logistic regression models predicting NRT engagement (*n* = 818).

| | (a) | | | |
|---|---|---|---|---|
| | **Intensive Arm** | | **Minimal Arm** | |
| | **Some Engagement** | **Complete Engagement** | **Some Engagement** | **Complete Engagement** |
| | **(1–7 Sessions)** | **(8 Sessions)** | **(1–2 Sessions)** | **(3 Sessions)** |
| | **Reference: 0 Sessions** | **Reference: 0 Sessions** | **Reference: 0 Sessions** | **Reference: 0 Sessions** |
| | **OR (95% CI)** | **OR (95% CI)** | **OR (95% CI)** | **OR (95% CI)** |
| | Demographics | | | |
| Age | *n/a* | *n/a* | | |
| 50–63 Years | | | 1.0 | 1.0 |
| 64–80 Years | | | 1.5 (0.78–2.8) | 1.9 (1.1–3.4) * |
| Gender | | | *n/a* | *n/a* |
| Male | 0.94 (0.48–1.8) | 1.6 (0.82–3.2) | | |
| Female | 1.0 | 1.0 | | |
| Race | | | *n/a* | *n/a* |
| White | 1.0 | 1.0 | | |
| Black | 1.1 (0.41–3.1) | 0.56 (0.18–1.8) | | |
| Marital Status | *n/a* | *n/a* | | |
| Married or in Marriage-like Relationship | | | 2.0 (1.1–3.8) * | 1.1 (0.62–2.0) |
| Not Married | | | 1.0 | 1.0 |
| Education Level | | | | |
| High School/GED or Less | 1.0 | 1.0 | 1.0 | 1.0 |
| Some College or Greater | 2.1 (1.1–4.0) * | 1.7 (0.85–3.3) | 2.1 (1.1–3.9) * | 1.9 (1.1–3.4) * |
| | Clinical Characteristics | | | |
| LDCT Screening | | | | |
| Baseline scan | 1.0 | 1.0 | 1.0 | 1.0 |
| Annual Scan | 2.1 (1.1–4.2) * | 1.9 (0.98–3.9) | 2.0 (1.04–3.8) * | 1.8 (0.99–3.2) |

**Table 2.** *Cont.*

| | (a) | | | |
|---|---|---|---|---|
| | Intensive Arm | | Minimal Arm | |
| | Some Engagement | Complete Engagement | Some Engagement | Complete Engagement |
| | (1–7 Sessions) | (8 Sessions) | (1–2 Sessions) | (3 Sessions) |
| | Reference: 0 Sessions | Reference: 0 Sessions | Reference: 0 Sessions | Reference: 0 Sessions |
| | OR (95% CI) | OR (95% CI) | OR (95% CI) | OR (95% CI) |
| Tobacco-Related Characteristics | | | | |
| Time to First Cigarette | *n*/a | *n*/a | | |
| Within 5 min | | | 1.2 (0.40–3.4) | 1.8 (0.68–4.6) |
| 6 to 30 min | | | 3.2 (1.03–9.9) * | 1.7 (9.57–5.3) |
| 31 to 60 min | | | 0.94(0.45–1.9) | 1.2 (0.62–2.3) |
| After 60 min | | | 1.0 | 1.0 |
| Confidence in Quitting | *n*/a | *n*/a | 1.0 (0.91–1.2) | 1.1 (0.99–1.3) |
| Psychological Variables | | | | |
| Comparative Risk (T1) | *n*/a | *n*/a | | |
| Lower risk | | | 1.0 | 1.0 |
| About the same | | | 1.0 (0.42–2.4) | 1.9 (0.83–4.2) |
| Higher risk | | | 1.9 (0.81–4.8) | 2.8 (1.2–6.6) * |
| (b) | | | | |
| | Intensive Arm | | Minimal Arm | |
| | Some NRT (up to 8 Weeks of NRT) Reference: No NRT | | Some NRT (2 Weeks of NRT) Reference: No NRT | |
| | OR (95% CI) | | OR (95% CI) | |
| Demographics | | | | |
| Education Level | | | *n*/a | |
| High School/GED or Less | 1.0 | | | |
| Some College or Greater | 1.6 (0.91–2.6) | | | |
| Tobacco-Related Characteristics | | | | |
| Pack Years | *n*/a | | | |
| 20–39 | | | 1.0 | |
| 40–49 | | | 1.5 (0.87–2.7) | |
| 50+ | | | 1.9 (1.1–3.5) * | |
| Cigarettes per day | *n*/a | | | |
| Less than 20 | | | 1.0 | |
| 20 or more | | | 1.2 (0.75–2.0) | |
| Time to First Cigarette | | | *n*/a | |
| Within 5 min | 2.0 (0.81–4.9) | | | |
| 6 to 30 min | 2.8 (1.3–6.2) * | | | |
| 31 to 60 min | 1.9 (0.83–4.1) | | | |
| After 60 min | 1.0 | | | |
| Motivation to Quit | 1.1 ǀ (0.97–1.2) | | *n*/a | |
| Psychological Variables | | | | |
| Comparative Risk (T1) | *n*/a | | | |
| Lower risk | | | 1.0 | |
| About the same | | | 1.8 (0.96–3.4) | |
| Higher risk | | | 2.7 (1.4–5.2) ** | |

\* $p < 0.05$; ** $p < 0.01$.

**Table 3.** Satisfaction with counseling and nicotine replacement therapy by study arm.

| | Intensive Arm (*n* = 312) | Minimal Arm (*n* = 306) | Total (*n* = 618) | *p*-Value |
|---|---|---|---|---|
| Counseling Feedback | | | | |
| Did you receive smoking cessation support from our project? | | | | |
| No | 14 (4.5) | 31 (10.2) | 45 (7.3) | |
| Yes | 297 (95.5) | 273 (89.8) | 570 (92.7) | 0.007 |
| Refused | 1 | 3 | 4 | |
| How satisfied were you with the counseling sessions that were conducted over the phone? (*n* = 570) | | | | |
| Not at all | 2 (0.7) | 8 (2.9) | 10 (1.8) | |
| A little satisfied | 10 (3.4) | 15 (5.5) | 25 (4.4) | |
| Somewhat satisfied | 35 (11.9) | 64 (23.4) | 99 (17.5) | |
| Very satisfied | 247 (84) | 186 (68.1) | 433 (76.4) | <0.001 |
| Refused | 3 | 0 | 3 | |

**Table 3.** *Cont.*

| | Intensive Arm (*n* = 312) | Minimal Arm (*n* = 306) | Total (*n* = 618) | *p*-Value |
|---|---|---|---|---|
| How much did the telephone counseling provided by this project help you to become more ready to quit smoking? | | | | |
| Not at all | 5 (1.7) | 16 (6) | 21 (3.8) | |
| A little | 15 (5.1) | 25 (9.4) | 40 (7.1) | |
| Somewhat | 67 (22.8) | 82 (30.8) | 149 (26.6) | |
| Very Much | 207 (70.4) | 143 (53.8) | 350 (62.5) | <0.001 |
| Refused | 3 | 5 | 8 | |
| Missing | 0 | 2 | 2 | |
| Would you have preferred more counseling calls, fewer calls, or it was the right number of calls? | | | | |
| Preferred Fewer Calls | 15 (5.2) | 7 (2.7) | 22 (4.0) | |
| It was the right amount of calls | 208 (71.5) | 156 (59.8) | 364 (65.9) | |
| Preferred more calls | 68 (23.4) | 98 (37.5) | 166 (30.1) | 0.001 |
| Refused | 6 | 9 | 15 | |
| Missing | 0 | 3 | 3 | |
| Please tell me about your preference for stop smoking counseling conducted over the phone vs. in person. | | | | |
| I prefer in person | 21 (7.2) | 24 (9.1) | 45 (8.1) | |
| Neutral/no preference | 131 (45) | 120 (45.3) | 251 (45.1) | |
| I prefer telephone counseling | 139 (47.8) | 121 (45.7) | 260 (46.8) | 0.68 |
| Refused | 6 | 6 | 12 | |
| Missing | 0 | 2 | 2 | |
| Nicotine Patch Feedback (*n* = 618) | | | | |
| Did you receive nicotine patches from our project? | | | | |
| No | 38 (12.3) | 54 (17.8) | 92 (15) | |
| Yes | 272 (87.7) | 249 (82.2) | 521 (85) | 0.056 |
| Refused | 2 | 3 | 5 | |
| Did you receive all of the patches that you requested? (*n* = 521) | | | | |
| No | 16 (6) | 12 (5.2) | 28 (5.6) | |
| Yes | 252 (94) | 217 (94.8) | 469 (94.4) | 0.725 |
| Refused | 1 | 1 | 2 | |
| Missing | 3 | 20 | 23 | |
| Did you use the nicotine patches? | | | | |
| No | 50 (18.4) | 84 (33.9) | 134 (25.8) | |
| Yes | 222 (81.6) | 164 (66.1) | 386 (74.2) | <0.001 |
| Missing | 0 | 2 | 2 | |
| How satisfied were you with the nicotine patches that you received? | | | | |
| Not at all satisfied | 15 (6.8) | 20 (12.3) | 35 (9.2) | |
| A Little satisfied | 15 (6.8) | 25 (15.4) | 40 (10.5) | |
| Somewhat satisfied | 61 (27.9) | 43 (26.5) | 104 (27.3) | |
| Very satisfied | 128 (58.4) | 74 (45.7) | 202 (53) | 0.006 |
| Refused | 8 | 7 | 15 | |
| How much did the nicotine patches provided by this project help you to become more ready to quit smoking? | | | | |
| Not at all | 25 (10.1) | 42 (20.2) | 67 (14.7) | |
| A Little | 32 (12.9) | 34 (16.3) | 66 (14.5) | |
| Somewhat | 53 (21.4) | 56 (26.9) | 109 (23.9) | |
| Very Much | 138 (55.6) | 76 (36.5) | 214 (46.9) | <0.001 |
| Refused | 17 | 29 | 46 | |
| Missing | 7 | 12 | 19 | |
| Did you purchase more patches to use, in addition to the ones we sent to you? | | | | |
| No | 230 (85.5) | 176 (72.4) | 406 (79.3) | |
| Yes | 39 (14.5) | 67 (27.6) | 106 (20.7) | <0.001 |
| Refused | 2 | 3 | 5 | |
| Missing | 2 | 2 | 4 | |

## 4. Discussion

The present analysis examined the predictors of cessation intervention engagement in a randomized trial in the lung screening setting. Overall, the majority of participants engaged in telephone counseling and received NRT during the intervention period. In the multivariable analyses, several demographic, tobacco-related, and psychological factors predicted engagement. In both the Intensive and Minimal interventions, more education and higher nicotine dependence were associated with greater intervention engagement, which is consistent with cessation trials in other settings [6,20]. A new finding specific to the lung screening setting was that coming in for an annual scan (compared to the baseline scan) predicted greater engagement within each study arm. The offer of smoking

cessation support at every screening exam is important, and understanding the influence of a patient's screening history may help to address barriers to engagement.

The results also indicated that participants' opinions about the interventions differed based on what was offered to them. The Intensive arm participants were more likely to report being very satisfied with the NRT, which could be due to the greater amount offered in this arm compared to the Minimal arm. This finding supports previous studies that have found that offering medication and NRT increased engagement [36–38]. Importantly, compared to the Minimal arm, Intensive arm participants were also more likely to report that the counseling and NRT provided by the trial helped them to become more ready to quit smoking. Participants who had more opportunities to engage in the intervention perceived the intervention to be more supportive of becoming ready to quit. The finding that Minimal arm participants were more likely to purchase additional NRT and less likely to report that the number of calls was just right (vs. the Intensive arm) suggests they may have been seeking additional support. It should be noted the present study did not offer other forms of smoking cessation medication as this was a pragmatic, remotely delivered intervention (telephone counseling and mailed NRT). However, future studies should assess methods of offering longer-term combined NRT or other pharmacotherapies in the lung screening context.

While some individual characteristics related to intervention engagement varied based on the intensity of the intervention provided, lower education, lower nicotine dependence, and undergoing a baseline scan were related to lower intervention engagement within both study arms. These findings can help inform implementation strategies to address barriers to integrating cessation services into the lung screening setting. For example, individuals who arrive for their first scan may be overwhelmed by the process or decide they do not need assistance with cessation and thus may be less likely to engage in cessation services. This finding underscores the importance of offering smoking cessation treatment at every scan and at multiple points in the screening continuum. Individuals' willingness to engage in treatment at the time of the annual scan may have to do with having been offered treatment previously. System changes that include creating EHR notifications of patients who are coming in for a repeat scan, but who have not quit smoking, or patients who have completed their first scan, but not yet initiated counseling, could be proactively contacted to engage them in treatment. Additionally, this finding underscores the importance of adherence to annual lung screening. Annual lung screening will not only maximize the benefit of the test itself, but it could also present new opportunities to engage patients in cessation treatment. Strategies are needed to automatically remind patients when it is time for their repeat scan and to prompt providers through the EHR. A second implementation strategy includes training teams (e.g., LCS staff, tobacco treatment specialists) on communication messages to support engagement in cessation programs. For example, patients with lower nicotine dependence may not recognize the importance of quitting or may underestimate the health effects of their tobacco use. Training teams to utilize targeted messages that communicate the health benefits of quitting in addition to getting screened are needed. This might include the use of plain language, motivational interviewing strategies, and teach-back methods for patients who have lower education or limited health literacy [39].

Study limitations include the lack of heterogeneity in the study sample, although it is reflective of the current lung screening population [40]. Subgroup analyses were limited by the small number of nonwhite participants. Future studies should examine levels of engagement by subgroups to inform strategies to promote engagement in cessation interventions. Second, as engagement in cessation trials has been defined in different ways, in this secondary data analysis, we explored various cut points based on these prior studies [6,18,21]. Finally, the impact of engagement on quit rates will be reported in the outcomes paper for the LSTH trial [33].

## 5. Conclusions

*What Is Next in the Implementation of Cessation Interventions at the Time of Lung Screening?*

The NCI SCALE clinical trials are investigating different methods of delivering smoking cessation treatment in the lung screening context. Importantly, these trials include individuals who are largely representative of the general lung screening-eligible population [41]. Taking the lessons learned from this and other SCALE trials that have evaluated reach, enrollment, and engagement, there are several considerations to inform the future integration of cessation in the lung screening context. Offering multiple accrual methods [7] and at multiple points in the LCS continuum [7,42] can assist with reach. Further, providing pharmacotherapy options promotes enrollment [7], and retention and treatment engagement differ on demographic, clinical, and psychological characteristics [13]. Low reach and engagement of patients in effective smoking cessation treatment at the time of lung screening will limit the impact that screening will have on tobacco-related mortality [43]. Following the publication of the SCALE trials, future work will need to test implementation strategies to determine how to reach and engage a large number of lung screening patients who currently smoke in order to maximize the reduction in mortality due to lung cancer.

**Author Contributions:** Conceptualization, R.M.W., E.E., L.S., J.G.P. and K.L.T.; methodology, E.E., L.S. and K.L.T.; formal analysis, R.M.W., E.E. and L.S.; investigation, E.E. and L.S.; resources, K.L.T.; data curation, E.E., L.S. and R.M.W.; writing—original draft preparation, R.M.W., E.E., L.S., J.G.P., J.W., M.W. and K.L.T.; writing—review and editing, J.G.P., J.W., M.W., T.L., G.L. and K.L.T.; visualization, L.S. and R.M.W.; supervision, K.L.T., T.L. and G.L.; project administration, L.S. and K.L.T.; funding acquisition, R.M.W. and K.L.T. All authors have read and agreed to the published version of the manuscript.

**Funding:** This research was funded by the National Cancer Institute at the National Institutes of Health (grant number R01CA207228) and the National Cancer Institute at the National Institutes of Health (grant number K99CA256515).

**Institutional Review Board Statement:** The study was conducted in accordance with the Declaration of Helsinki and approved by the Institutional Review Board of Georgetown University Medical Center (2016-0651) and by the Lahey Hospital and Medical Center IRB (2017-022).

**Informed Consent Statement:** Informed consent was obtained from all subjects involved in the study.

**Data Availability Statement:** The data underlying this article will be shared on reasonable request to the corresponding author.

**Acknowledgments:** The authors are grateful to all of the participants who contributed their time. To the lung cancer screening sites: Georgetown University Medical Center, Washington, DC, Lahey Hospital & Medical Center, Burlington, MA, Hackensack University Medical Center, Hackensack, NJ, Baptist Hospital of Miami, Miami, FL, Hartford Hospital, Hartford, CT, UnityPoint Health, Des Moines, IA, MedStar Shah Medical Group, Ft. Washington, MD, Anne Arundel Medical Center, Annapolis, MD. To the site investigators: Eric Anderson (GU), Juan Batlle (Baptist Health), Harry Harper (HUMC), Andrea McKee (Lahey), Brady McKee (Lahey), Vicky Parikh (MedStar Shah), Ellen Dornelas (Hartford), Judith Howell (Unity Point), Maria M. Geronimo (AAMC). To our consultants: David B. Abrams, Jennifer Frey, Raymond S. Niaura, and Cassandra Stanton. To our tobacco treatment specialists: Claudia Campos, Marisa Cordon, Danielle Deros, Jennifer Stephens, and Shelby Fallon. To our project coordinators, interviewers, and the entire LSTH study team: Emily Kim, Jen-Yuan Kao, Daisy Dunlap, Sarah Hutchison, Julia Friberg, Lisa Charles, and Ryan Anderson.

**Conflicts of Interest:** The authors declare no conflict of interest. The funders had no role in the design of the study; in the collection, analyses, or interpretation of data; in the writing of the manuscript; or in the decision to publish the results.

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
