# Peer review of "Engaging Patients in Smoking Cessation Treatment within the Lung Cancer Screening Setting: Lessons Learned from an NCI SCALE Trial"

_curroncol, doi:10.3390/curroncol29040180_

Round 1

Reviewer 1 Report

This is an important and well written paper. I have the following comments which may improve the presentation and discussion.

Only 8% preferred in person cessation. What was the engagement of those who preferred in person to telephone, was there a difference in the demographics of those who preferred in person.

Authors should discuss the observation that since repeat lung cancer scans were associated with increased engagement, can systems to promote and remind patients about rescanning leat to increased cessation engagement.

There was no mention of insurance status and SDOH or area deprivation in patients. Were these collected, if not why not, and what were the results if any information was collected.

Race information white vs black was collected, but results in white vs black were not discussed. The engagement in whites was higher than blacks, significant but not discussed. This aspect of results should be discussed, since it is so important in the context of worse lung cancer results in blacks, and authors should discuss if the associations of demographics are different in blacks vs whites. should the recommendations be different in whites vs blacks. 

Author Response

Reviewer 1: This is an important and well written paper. I have the following comments which may improve the presentation and discussion.

1. Only 8% preferred in person cessation. What was the engagement of those who preferred in person to telephone, was there a difference in the demographics of those who preferred in person.

Response: Thank you for your review and feedback. Due to the small number of participants who preferred in person counseling, we were unable to to look at the level of counseling engagement by this variable (resulted in cells with 0 cases). There were no differences in the demographics of those who preferred in person counseling.

2. Authors should discuss the observation that since repeat lung cancer scans were associated with increased engagement, can systems to promote and remind patients about rescanning lead to increased cessation engagement.

Response: Thank you for this suggestion. We have added this point to the Discussion (page 13).

3. There was no mention of insurance status and SDOH or area deprivation in patients. Were these collected, if not why not, and what were the results if any information was collected.

Response: Thank you for this comment. We collected health insurance and more than 95% of the sample was insured. Instead, we did include education as a measure of socioeconomic status. When stratifying the data by study arm and the engagement variables, it resulted in cells with 0 cases (for example, among the Intensive arm there were no participants who were uninsured and who did not receive NRT). Given these small cell sizes, we did not include insurance in the present analysis. We have included a comment about this in the Measures Section (page 4).

4. Race information white vs black was collected, but results in white vs black were not discussed. The engagement in whites was higher than blacks, significant but not discussed. This aspect of results should be discussed, since it is so important in the context of worse lung cancer results in blacks, and authors should discuss if the associations of demographics are different in blacks vs whites. should the recommendations be different in whites vs blacks.

Response: We agree this is an important point given the lung cancer disparities affecting Black/African American individuals. Although race was associated with counseling engagement at the univariate level, when entered into multivariable analyses, it was no longer significant. While we did detect univariate differences, we were reluctant to include this in the interpretation of the findings given the small sample size. We have added a call for additional research in this area to the Discussion section (page 14).

Reviewer 2 Report

This is an excellent paper on a very important topic.

I just have a couple of points that need to be addressed:

  1. In a couple of places including the abstract it talks about patients requesting NRT but in the methods it states they were given NRT. This needs to be qualified because the suggestion that patients are left to request NRT must surely be wrong.
  2. Was any other form of smoking cessation medication considered? This should be addressed in the discussion. Varenicline is now recommended as a first line therapy to assist smoking cessation in some jurisdictions including my own, with NRT second line. There are other therapies also incuding nortriptyline for certain patients.
  3. There is a presumable typographical error in line 163 where it is stated "receive nicotine patches from patches".

Author Response

Reviewer 2:  This is an excellent paper on a very important topic. I just have a couple of points that need to be addressed:

1. In a couple of places including the abstract it talks about patients requesting NRT but in the methods it states they were given NRT. This needs to be qualified because the suggestion that patients are left to request NRT must surely be wrong.

Response: Thank you for your review and for this important comment. All participants were offered NRT and we mailed it to those who wished to use it. To reduce confusion, we have updated the manuscript to refer to this variable as NRT mailed by the LSTH study team.

2. Was any other form of smoking cessation medication considered? This should be addressed in the discussion. Varenicline is now recommended as a first line therapy to assist smoking cessation in some jurisdictions including my own, with NRT second line. There are other therapies also including nortriptyline for certain patients.

Response: Other forms of smoking cessation medication were not included in the current study as this was intended to be a remotely delivered intervention with telephone counseling and over the counter medication that could be mailed to patients without a prescription.  We were testing an intervention that was pragmatic and could be implemented in other lung screening settings. However, we agree that additional research is needed to assess methods of offering longer-term combined NRT or other pharmacotherapies in the lung cancer screening context. We have addressed this point in the Discussion section (page 13).

3.There is a presumable typographical error in line 163 where it is stated "receive nicotine patches from patches".

Response: Thank you for making note of this typographical error. It should read ‘Did you receive nicotine patches from our project?’ which we have updated in the revised manuscript (Section 2.3 Measures, page 4).